# Effect of Praziquantel Treatment on the Nutritional Status of Children Infected with *Schistosoma haematobium*

**DOI:** 10.3390/pathogens14020123

**Published:** 2025-01-29

**Authors:** Louis Fok, Hongying Daisy Dai, David M. Brett-Major, Abebe Animut, Berhanu Erko, John Linville, Yohannes Negash, Abraham Degarege

**Affiliations:** 1Department of Epidemiology, College of Public Health, University of Nebraska Medical Center, Omaha, NE 68198, USA; louis.fok@unmc.edu (L.F.); david.brettmajor@unmc.edu (D.M.B.-M.); 2Department of Biostatistics, University of Nebraska Medical Center, Omaha, NE 68198, USA; daisy.dai@unmc.edu; 3Division of Infectious Diseases, College of Medicine, University of Nebraska Medical Center, Omaha, NE 68198, USA; 4Aklilu Lemma Institute of Pathobiology, Addis Ababa University, Addis Ababa P.O. Box 1176, Ethiopia; abebe.animut@aau.edu.et (A.A.); berhanu.erko@aau.edu.et (B.E.); yohannes.negash@aau.edu.et (Y.N.); 5Department of Environmental, Agricultural & Occupational Health, University of Nebraska Medical Center, Omaha, NE 68198, USA; john.linville@volunteer.unmc.edu

**Keywords:** praziquantel, urogenital schistosomiasis, *Schistosoma haematobium*, nutrition

## Abstract

This quasi-experimental trial examined the relationship between *Schistosoma haematobium* infection and nutritional status, and the impact of single dose praziquantel (PZQ) therapy on undernutrition. A total of 353 children were examined, 112 of which were infected with *S. haematobium* and treated with PZQ. Children’s heights, weights, and mid-upper arm circumferences (MUAC) were measured at baseline and one month post-treatment. Infected children had significantly smaller mean BMI-for-age z-scores (BAZ) (−1.16 vs. 0.11, *p* < 0.01) and weight-for-age z-scores (WAZ) (−0.61 vs. −0.31, *p* = 0.03) than the uninfected ones at baseline. *S. haematobium* infection was associated with underweight (adjusted OR: 1.76, 95% CI: 1.63–1.90). One month after treatment, BAZ, WAZ, height for age z-scores (HAZ), and MUAC scores were comparable between treated and control children. However, there was a significant decrease in the prevalence of underweight among treated children, while no significant change was observed in the control group one month post-treatment. In conclusion, children infected with *S. haematobium* are likely to suffer from undernutrition; however, single dose PZQ therapy may not improve their nutritional status within one month of treatment. Future studies could have longer follow-up periods to better estimate the drug’s effect on nutrition.

## 1. Introduction

Urogenital schistosomiasis, caused by the helminth *Schistosoma haematobium*, remains a significant public health concern, particularly in sub-Saharan Africa. It is a major cause of morbidity, mortality, and disability worldwide [1], disproportionately affecting children due to their frequent contact with infected sources of water [2]. Indeed, the disease is infamous for the amount of disability it causes—undernutrition being one of the prime outcomes of interest. In addition to causing major damage to the urinary tract in 70% of infected children [3], it is well-known that the infection negatively impacts child growth by causing anemia and stunting [4] as both the helminths and their eggs consume the host’s (human’s) metabolites [5] and divert blood away from organs [3]. The World Health Organization claims that schistosomiasis-related undernutrition is “usually reversible with treatment” [4]. Moreover, some researchers have posited that treating the disease is “likely to improve child growth, appetite, physical fitness and activity levels” [6]. However, the evidence on this topic is surprisingly sparse and somewhat contradictory.

Despite the widespread use of praziquantel (PZQ) as an antischistosomal drug, the literature lacks comprehensive data on its impact on the nutritional status of infected children. A study reported thicker skin and higher hemoglobin levels among children infected with *S. japonicum* in the Philippines one month after PZQ treatment [7]. Another study in Tanzania documented improved hemoglobin levels and increased prevalence of wasting but lack of changes in the prevalence of stunting among children infected with *S. mansoni* eight months after PZQ treatment [8]. On the other hand, a study from Kenya showed improved appetite and physical fitness among children infected with *S. haematobium* and hookworms after treatment with praziquantel and/or metrifonate [9]. A clinical trial that involved Kenyan children discovered that, compared to a placebo, single dose praziquantel was successful at increasing weight, weight-for-age, weight-for-height, arm circumference, and skin thickness eight months post-treatment [10]. Likewise, PZQ treatment has been associated with increases in weight-for-age z-scores and BMI-for-age z-scores [11] and decreases in anemia [12] among *S. mansoni*-infected Ethiopian school children.

The preponderance of evidence suggests that praziquantel may have a beneficial effect on children’s nutritional status within the medium term (half a year to a year) [13]. To date, few studies have empirically and comprehensively measured the effect of PZQ therapy on the nutritional status of children infected with *S. haematobium*. Determining the effects of PZQ treatment on the nutritional status of *S. haematobium*-infected children is important given that the species primarily infects the urogenital tract and not the intestine, unlike all the other species. Thus, uncovering the effect of PZQ treatment on the nutritional status of *S. haematobium-*infected children would enable us to better gauge such species’ impact on children’s physical development. In addition, studies are scarce on this topic within the Ethiopian context, and none have examined the time needed to ameliorate *S. haematobium*-associated undernutrition. This study, therefore, provides evidence of how quickly undernutrition can be reversed among children with *S. haematobium* treated with PZQ. It is critical to understand how fast undernutrition can be addressed in children infected with *S. haematobium* as prolonged undernutrition increases the likelihood of permanent developmental deficits and reduces the chance of achieving developmental “catch-up” later in life [14,15]. Hence, it is necessary to fill this knowledge gap with real-world evidence. Therefore, to confirm past research on this topic and to replicate it on children specifically infected with *S. haematobium*, in Ethiopia, and within a smaller timeframe, this study assessed the nutritional status of *S. haematobium*-infected children and examined whether praziquantel therapy improves anthropometric nutritional parameters within a one month period of drug administration.

## 2. Materials and Methods

### 2.1. Study Design

To examine the relationship between *S. haematobium* infection and the nutritional status of children at baseline and one month post-treatment with PZQ, a cross-sectional study of baseline characteristics was conducted. To evaluate the effect of praziquantel therapy on children’s nutritional status, a quasi-experimental pilot study with a nonequivalent control group was performed among children residing in Afar and Gambella, Ethiopia. The research protocol was part of a larger trial conducted in the Afar, Gambella, and Benishangul-Gumuz regional states of the country to evaluate the use of pooled testing as a cost-effective strategy for diagnosing *S. haematobium* [16]. As such, the sample size was pre-determined.

### 2.2. Study Participants and Sites

In July 2023, 572 children between the ages of 5 and 15 years residing in villages located along the Middle Awash Valley of Afar Region, Ethiopia were recruited for this study; in July 2024, a further 405 children of the same age range residing in villages of the Gambella Region were recruited. Villages were selected based on prior studies [17,18] where a higher prevalence of *S. haematobium* infections was reported, in collaboration with local officials and guides who took accessibility into consideration. Children were only allowed to participate if they resided in a village that had not received mass praziquantel administration in the last ninety days and informed consent was obtained from them and their parents.

### 2.3. Data Collection

Upon admission, each child’s height (in centimeters), weight (in kilograms), mid-upper arm circumference (MUAC, in centimeters), sex, age, and village were recorded. The child was then directed to urinate at least 100 mL of urine into a plastic jar. To expedite treatment in light of the sheer volume of children being analyzed, those whose urine samples tested positive for hematuria were provided with praziquantel (Bermoxel by Medochemie, Limassol, Cyprus) at a single dose of 40 mg/kg. This was completed given that research has confirmed that the rapid testing of urine for hematuria has been found to be an accurate proxy for detecting *S. haematobium* infection [19]. The instructions from the manufacturer were followed to check for the presence of hematuria using Combur 10 Test reagent strips (Roche Diagnostics, Rotkreuz, Switzerland). In brief, the test strip was immersed in the urine sample for about two seconds, then removed and left on the bench for approximately one minute. The blood area on the test strip was then compared to the corresponding reference color fields. The strip test results were interpreted and recorded as zero (negative), ± (trace), + (weak), ++ (moderate), and +++ (strong), indicating the presence of erythrocytes per μL of urine.

All samples underwent confirmatory testing for *S. haematobium* eggs using standard urine filtration microscopy in accordance with the parent study [16], by which a 10 mL aliquot of each urine sample was filtered through a polycarbonate membrane and then analyzed for *S. haematobium* eggs under the microscope in the lab, the results of which were used to group children into treated (i.e., urine samples tested positive for *S. haematobium* egg and treated with PZQ) and control (i.e., urine samples tested negative for *S. haematobium* egg and untreated with PZQ) groups. Efforts were made to ensure that children who tested negative for hematuria but who tested positive using urine filtration microscopy were promptly treated with PZQ. Approximately one month after PZQ treatment of *S. haematobium*-infected children, the research team revisited the villages, recalled all the children, and had their height, weight, and mid-upper arm circumference re-measured. Of the 245 children who were confirmed positive for *S. haematobium* eggs at baseline, 52 were excluded from the study because they tested negative for hematuria in the field and were therefore not treated until later. Likewise, of the 721 children who were confirmed negative for *S. haematobium* eggs, 91 were excluded because they tested positive for hematuria in the field and were therefore given praziquantel. See Figure 1 for a description of how the sample sizes for treatment and control groups were reached.

### 2.4. Analysis

Of the 977 recruited children, 353 were successfully followed up for re-measurement. The results of this study were analyzed entirely on an intent-to-treat basis.

#### 2.4.1. Determining Nutritional Status

The participants’ heights, weights, and ages were inputted into the World Health Organization’s AnthroPlus software (Version 1.0.4, Geneva, Switzerland) to categorize the children’s underweight, wasting, and stunting status [15]. Children whose BMI-for-age z-values (BAZ) were −2 or lower were categorized as “underweight;” those whose weight-for-age z-values (WAZ) were −2 or lower were categorized as “wasted;” and those whose height-for-age z-values (HAZ) were −2 or lower were categorized as “stunted”.

#### 2.4.2. Association of *S. haematobium* Infection with Undernutrition at Baseline

Three generalized estimating equation models (GEE) were employed to examine the relationship between *S. haematobium* infection and undernutrition. In these three GEE models, the outcome variables were underweight (yes/no), stunting (yes/no), and wasting (yes/no), and the independent variable was *S. haematobium* infection status (yes/no). Furthermore, four linear regression models were used to estimate the changes in the mean MUAC, BAZ, WAZ, and HAZ values at baseline as *S. haematobium* infection status (yes/no) changes. In all seven models, the region (Afar vs. Gambella) where the children live was included/controlled as the cluster variable [20]. Age and sex were not included in the models, given that the nutritional parameters already took them into account. For all models, 95% confidence intervals were reported.

#### 2.4.3. Effect of Praziquantel Treatment on Nutritional Status

Multiple statistical tests were employed to measure the effect of praziquantel treatment on nutritional status. First, paired samples t-tests were performed to compare the changes in the means of MUAC, BAZ, WAZ, and HAZ scores pre- and post-treatment between children who were infected with *S. haematobium* and treated with praziquantel and those who were uninfected with the parasite at baseline. Then generalized linear mixed models (GLMM) were employed to compare the individual level changes in the means of MUAC, BAZ, WAZ, and HAZ scores pre- and post-treatment between children who were infected with *S. haematobium* and treated with praziquantel and those who were uninfected with the parasite at baseline after controlling for the villages where they live. The GLMM model for MUAC also adjusted for age and sex given that such variables are not taken into account for MUAC, unlike BAZ, WAZ, and HAZ. In all four GLMM models, ID was used as the random variable.

Similarly, McNemar’s chi-square test was performed to compare the changes in the prevalence of underweight, wasting, and stunting before and after treatment between children infected and uninfected with *S. haematobium*. Then, GEE models were executed to estimate the population-level change in the prevalence of underweight, wasting, and stunting before and after treatment between children infected and uninfected with *S. haematobium*. Village was treated as a cluster variable. Statistical significance was set at an alpha level of 0.05. All statistical analyses were executed using SAS version 9.4 (SAS Institute, Cary, NC, USA).

### 2.5. Ethical Approval

Ethical approval was obtained from the Institutional Review Boards of the University of Nebraska Medical Center (IRB# 908-19-EP) and Aklilu Lemma Institute of Pathobiology, Addis Ababa University (Ref no. ALIPB-IRB/10/2012/20). This study included only children who agreed to participate and for whom parental written consent was obtained.

## 3. Results

### 3.1. Baseline Demographic and Nutritional Characteristics of Study Participants

Of the 353 children included in the study, 32% were *S. haematobium* positive and treated with praziquantel (Table 1). The participants were almost evenly divided between females and males. Females and males were also evenly divided between treatment and control groups. A majority of the study participants (79%) were young (aged from five to ten years). Out of the 112 children in the treatment group, 35 (31%) were aged from 11 to 15 years, whilst 38 (16%) out of the 241 children in the control group were in this age group. Of all the children included in the study, 72% of the ones from Gambella tested positive for *S. haematobium* infection compared to 18% of those from Afar (*p* < 0.01).

### 3.2. Relationship Between S. haematobium Infection and Undernutrition

At baseline, children infected with *S. haematobium* had a lower BAZ (−1.16 vs. 0.11, *p* < 0.01), and WAZ (−0.61 vs. −0.31, *p* = 0.03) score compared to uninfected children (Table 1). The prevalence of underweight (25% vs. 4%, *p* < 0.01) and wasting (13% vs. 3%, *p* < 0.01) was also greater among the infected than the uninfected children at baseline. However, the infected children had greater mean MUAC scores (17.18 vs. 16.55, *p* < 0.01) and HAZ scores (−0.35 vs. −0.87, *p* < 0.01) than the uninfected ones. There was no statistically significant difference in the prevalence of stunting between infected and uninfected children (*p* = 0.07).

After controlling for region, children who were infected with *S. haematobium* had 76% greater odds of being underweight compared to uninfected children (aOR: 1.76, 95% CI: 1.63–1.90) (Table 2). Infection status did not appear to have a significant effect on any of the other nutritional parameters at an alpha level of 0.05 in the adjusted model. Nonetheless, a child living in Afar was less likely to be underweight (aOR: 0.03, 95% CI: 0.03–0.03). Likewise, a child living in Afar was also less likely to be wasted (aOR: 0.17, 95% CI: 0.07–0.38).

### 3.3. Impact of Praziquantel Therapy on Nutritional Status

Crude analysis showed an increase in mean WAZ scores among children infected with *S. haematobium* after being treated with praziquantel (+14.8%, *p* = 0.03) (Table 3). On the other hand, children who were not treated experienced a decrease in mean BAZ scores (−181.8%, *p* < 0.01) but an increased mean HAZ scores (+11.5%, *p* = 0.04). Though the difference was not significant, the prevalence of wasting decreased among treated children one month after PZQ therapy (−40%, *p* = 0.10) but increased among the controls (+42.9%, *p* = 0.08). The decrease in the prevalence of underweight among children treated with praziquantel compared to the control children was marginally significant (*p* = 0.05) after clustering for village. None of the other changes in the nutritional parameters showed significant change one month after praziquantel treatment in the adjusted/clustered models.

Children who received praziquantel had 47% lower odds of being underweight one month following treatment compared to the non-treated (OR = 0.53, 95% CI: 0.28–1.00) (Table 4). Likewise, the treated children had 34% lower odds of being stunted one month following treatment compared to the controls (OR = 0.66, 95% CI: 0.35–1.24), although this was not statistically significant (*p* = 0.20). Paradoxically, treated children appeared to have greater odds of being wasted one month after treatment compared to the control children (OR = 5.89, 95% CIL 0.28–124.81), however, the difference was not statistically significant (*p* = 0.26) and the confidence interval was too wide to draw any conclusion from this estimate.

## 4. Discussion

This study evaluated the relationship between *S. haematobium* infection and undernutrition in children and the changes in nutritional status following PZQ treatment. It demonstrated that children infected with *S. haematobium* have significantly lower BAZ and WAZ scores than uninfected children at baseline. After adjusting for region, *S. haematobium* infection was also significantly associated with increased odds of being underweight. PZQ therapy had a marginally statistically significant effect on reducing the prevalence of underweight. However, the changes in the mean MUAC, BAZ, WAZ, HAZ, and the prevalences of wasting and stunting were non-significant one month after PZQ therapy.

This study confirmed the findings from past research that demonstrated an association between *S. haematobium* infection and undernutrition [4,21]. At baseline, infected children had lower mean BAZ and WAZ scores than uninfected children. Moreover, infected children had a greater prevalence of wasting and underweight than uninfected children, and infection is a statistically significant predictor of being underweight. These findings lend credence to the notion that undernutrition may be a major symptom of *S. haematobium* infection, whether it be caused by the parasite consuming the host’s metabolites [5], through the inducement of abdominal pain and diarrhea [4], or the alteration of the child’s metabolism [21] (or some combination thereof).

Although the difference was not statistically significant, the treated children benefitted from a 14.8% increase in their mean weight-for-age z-scores, whilst the control children suffered a decline of 22.6%. Moreover, the treated children experienced a 40% decrease in wasting prevalence whilst control children saw a 42.9% increase. These results align with a meta-analysis that found schistosomiasis to be associated with wasting [22]. Yet, our cross-sectional analysis found no statistically significant relationship between *S. haematobium* infection and wasting, which may have been due to the relatively small sample size. Future research should seek to clarify the interplay between schistosomiasis, praziquantel therapy, and wasting status.

In terms of public health, this study suggests that single-dose praziquantel monotherapy is unlikely to significantly reverse *S. haematobium*-associated undernutrition in the short term among infected children in an endemic setting. Whilst these results may not align with our hypothesis, it should be cautioned that they only present the effect of praziquantel treatment one month after treatment. One month of follow-up time is likely insufficient to detect any significant change in the children’s physical measurements and falls well short of the follow-up time that was used in previous trials on the use of praziquantel to improve nutritional status [7,23]. This underscores the need to conduct future studies with longer follow-up periods and may be with repeated PZQ treatment. After all, the effect that nutritional intake has on the body develops over the course of longer time frames [24].

The main advantage of this study was the presence of a control group to evaluate the changes in the nutrition status due to PZQ treatment. Nonetheless, because of the context of this research setting as well as resource limitations, we were unable to collect richer data that would have enabled us to control for other biasing factors, including a history of past infection, co-infection with other parasites and soil-transmitted helminths, exposure to bodies of water during the follow-up period which may cause rapid reinfection, and other social determinants of undernutrition, such as food insecurity, availability and access to health foods, and financial insecurity. Likewise, we were unable to measure more facets of nutritional status, including those that can only be measured using clinical/laboratory testing, including blood iron levels, blood glucose levels, proteinuria, and skin thickness, among others, as were done in other studies [7,23]. Given that the praziquantel was provided to parents with instructions for administration to their infected children and that during the follow-up survey, the researchers were unable to ascertain drug consumption due to logistical constraints, only an intent-to-treat analysis was performed. As such, the nondifferential misclassification that may have resulted would have likely biased the results of this study toward the null.

Moreover, it should be noted that this study experienced a large loss to follow-up as 42% of the experimental group and 62% of the control group failed to show up in the follow-up survey. As the participants of this study consisted primarily of nomadic Afar and displaced persons living in the war-torn region of Gambella, successfully following up with the children was challenging. Nonetheless, there was a major difference in follow-up rates between the two groups, which poses a serious problem for this study’s internal validity [25,26]. Prior studies have identified various factors that are strongly associated with loss of follow-up among Ethiopian children, including living in rural areas—which describes this study population well—and poor drug adherence [27]. A comparison of those children who were and were not lost to follow-up is summarized in Appendix A.

In addition, it must be noted that different personnel performed pre- and post-treatment measurements. This might have introduced bias in measurements at the two time points due to differing techniques and may partially account for why some post-treatment measurements were implausibly different from the pre-treatment ones (in particular, the MUAC scores for the control group). It should be noted that since an analog scale was used, the children’s weights were read in increments of 1 kg. This study was, therefore, likely insensitive to minute changes in the children’s weights. Furthermore, because of the cross-sectional nature of this baseline survey, it cannot be definitively proven that the undernutrition observed was caused by *S. haematobium* infection. Finally, this study’s sample size was 353, which, despite being larger than that of a previous trial [7], is still relatively small and lacks statistical power. Preliminary calculations estimated that the ideal sample size would be 684. However, because this study was a spinoff of a larger research venture, our recruitment fell well short of that estimate. Future studies should be performed with a greater number of participants.

Prior research has uncovered those other variables, like village, age, and sex, are significantly associated with undernutrition [28,29]. Echoing those findings, this study has uncovered that region is significantly associated with underweight, stunting, and wasting status, as well as MUAC and BAZ. It could thus be posited that the environment in which a child grows up may have a stronger effect on his or her nutritional status than the consumption of an anthelminthic. After all, previous research has found that administering antimicrobials to African children does not improve growth [30,31]. Public health efforts that aim to improve childhood nutrition in Africa may be put to better use by focusing on interventions that seek to affect how social and environmental dynamics impact a child’s nutrition rather than employing pharmaceutical interventions.

## 5. Conclusions

Urogenital schistosomiasis may cause underweight and wasting in school-aged children. However, treating the infection with a single dose of praziquantel does not appear to significantly improve anthropometric nutritional parameters within one month of treatment. Future studies should consider monitoring children over longer periods, incorporating repeated praziquantel treatments to better understand the potential for reversing *S. haematobium*-related undernutrition. Controlling for other social and environmental factors influencing nutrition will also be essential during such studies.

## Figures and Tables

**Figure 1 pathogens-14-00123-f001:**
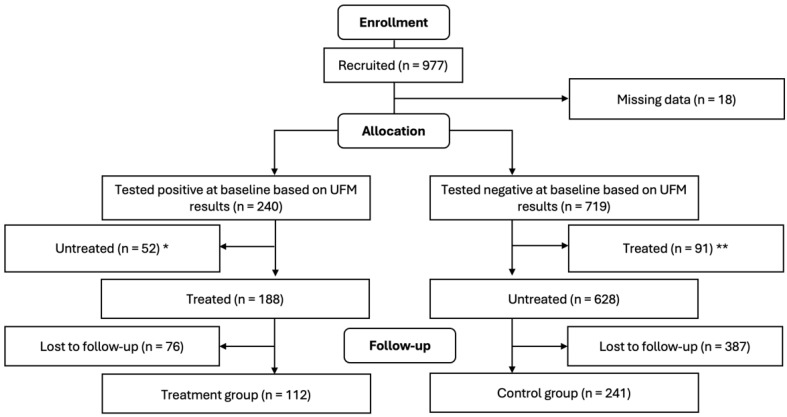
Flow diagram of study participants (treatment was single dose PZQ 40 mg/kg). * These 52 children were untreated and excluded from this study because they tested negative for hematuria in the field using urine reagent strips but were eventually found positive in the lab using UFM. ** These 91 children were treated and excluded from this study because they tested positive for hematuria in the field using urine reagent strips but were eventually found negative in the lab using UFM.

**Table 1 pathogens-14-00123-t001:** Baseline demographic and nutritional characteristics of the study participants.

		Treatment Group		Total	*p*
		Praziquantel (Infected)	Control (Uninfected)		
Sex	Female	51 (31%)	114 (69%)	165	0.82
	Male	61 (32%)	127 (68%)	188	
Age	5–10	77 (27%)	203 (73%)	280	<0.01
	11–15	35 (48%)	38 (52%)	73	
Village	Andada	5 (14%)	30 (86%)	35	^a^
	Abaro	18 (100%)	0 (0%)	18	
	Buri	14 (22%)	51 (78%)	65	
	Erinbirta	1 (11%)	8 (89%)	9	
	Gabole	8 (44%)	10 (56%)	18	
	Haledebe	11 (22%)	40 (78%)	51	
	Iore	1 (17%)	5 (83%)	6	
	Kalat	3 (13%)	20 (87%)	23	
	Kusara	5 (0%)	34 (100%)	39	
	Mender17	26 (52%)	24 (48%)	50	
	Office	0 (0%)	6 (100%)	6	
	Rebada	1 (7%)	13 (93%)	14	
	Tegni	19 (100%)	0 (0%)	19	
Region	Afar	49 (18%)	217 (82%)	266	<0.01
	Gambella	63 (72%)	24 (28%)	87	
Underweight	Yes	28 (74%)	10 (26%)	38	<0.01
	No	84 (27%)	231 (73%)	315	
Wasted *	Yes	10 (59%)	7 (41%)	17	<0.01
	No	67 (25%)	196 (75%)	263	
Stunted	Yes	10 (20%)	39 (80%)	49	0.07
	No	102 (34%)	202 (66%)	304	
Mean MUAC		17.18	16.55		<0.01
Mean BAZ		−1.16	0.11		<0.01
Mean WAZ *		−0.61	−0.31		0.03
Mean HAZ		−0.35	−0.87		<0.01
Total		112 (32%)	241 (68%)	353	

* Calculations for WAZ and wasting status only included children 10 years of age or younger; ^a^
*p*-value could not be calculated because the model failed to converge.

**Table 2 pathogens-14-00123-t002:** Predictors of baseline nutritional status among children residing in Afar and Gambella, July 2023 & July 2024.

		Underweight ^1^ aOR (95% CI)	Wasted ^1^ aOR (95% CI)	Stunted ^1^ aOR (95% CI)	MUAC ^2^ β (95% CI)	BAZ ^2^ β (95% CI)	WAZ ^2^ β (95% CI)	HAZ ^2^ β (95% CI)
Infection status	Infected	1.76 (1.63–1.90)	1.78 (0.38–8.38)	0.98 (0.81–1.20)	−0.10 (−0.39–0.19)	−0.44 (−1.58–0.70)	−0.09 (−5.00–4.83)	−0.03 (−2.69–2.63)
	Uninfected	Reference	Reference	Reference	Reference	Reference	Reference	Reference
Region	Afar	0.03 (0.03–0.03)	0.17 (0.07–0.38)	9.04 (8.13–10.06)	−1.60 (−1.76–−1.44)	1.79 (1.18–2.41)	0.48 (−2.06–3.02)	−1.18 (−2.62–0.25)
	Gambella	Reference	Reference	Reference	Reference	Reference	Reference	Reference

^1^ GEE model with region as a cluster variable; ^2^ linear regression models with region as a cluster variable.

**Table 3 pathogens-14-00123-t003:** Nutritional status among children before and after praziquantel treatment.

	Treatment Received								*p* ^2^
	Praziquantel				Control				
	Pre	Post	Δ%	*p* ^1^	Pre	Post	Δ%	*p* ^1^	
MUAC	17.19	17.16	−0.2%	0.68 ^a^	16.55	23.93	44.6%	0.33 ^a^	0.55 ^c^
BAZ	−1.16	−1.10	5.2%	0.48 ^a^	0.11	−0.09	−181.8%	<0.01 ^a^	0.19 ^d^
WAZ	−0.61	−0.52	14.8%	0.03 ^a^	−0.31	−0.38	−22.6%	0.07 ^a^	0.23 ^d^
HAZ	−0.35	−0.35	0.0%	0.98 ^a^	−0.87	−0.77	11.5%	0.04 ^a^	0.11 ^d^
Underweight	25.00%	21.43%	−14.3%	0.28 ^b^	4.15%	5.39%	30.0%	0.41 ^b^	0.05 ^e^
Wasted	12.99%	7.79%	−40.0%	0.10 ^b^	3.45%	4.93%	42.9%	0.08 ^b^	0.26 ^e^
Stunted	8.93%	10.71%	20.0%	0.41 ^b^	16.18%	13.69%	−15.4%	0.18 ^b^	0.20 ^e^

^1^ *p*-value to detect a difference in pre- and post-treatment nutritional measurements within the same treatment or control group; ^2^ *p*-value to detect a difference in the change in nutritional measurements pre- and post-treatment between praziquantel and control groups; ^a^ *p*-value was obtained by paired-samples *t*-test; ^b^ *p*-value was obtained by McNemar’s chi-square test; ^c^ *p*-value was obtained by generalized linear mixed model analysis adjusting for age, sex, and village; ^d^ *p*-value was obtained by generalized linear mixed model analysis adjusting for village only; ^e^ *p*-value was obtained by generalized estimating equations analysis treating village as a cluster variable.

**Table 4 pathogens-14-00123-t004:** The odds ratios for dichotomous nutritional parameters in treated versus non-treated children residing in Afar and Gambella one month following praziquantel therapy, July 2023 & July 2024.

Outcome Measure	Comparison	OR (95% CI)	*p*-Value
Being underweight	Praziquantel vs. control group	0.53 (0.28–1.00)	0.05
Being wasted	Praziquantel vs. control group	5.89 (0.28–124.81)	0.26
Being stunted	Praziquantel vs. control group	0.66 (0.35–1.24)	0.20

## Data Availability

The data presented in this study are available on request from the corresponding author. The data are not publicly available due to privacy/ethical issues.

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
