# Peer review of "Effect of Praziquantel Treatment on the Nutritional Status of Children Infected with Schistosoma haematobium"

_pathogens, 2025, doi:10.3390/pathogens14020123_

Round 1

Reviewer 1 Report

Comments and Suggestions for Authors

The manuscript highlighted an important aspect of Schistosoma haematobium infection, treatment of infected children with praziquantel (PZQ), and observation of improvement in nutritional status. The study evaluated if single dose of PZQ is able to improve nutritional status among infected children compared to uninfected children. This is important aspect of Schistosomiasis infection that needs further investigation. However, the manuscript has drawbacks that needs improvement.

Introduction:

Line 67-70: This sentence needs rewriting or breaking it into two sentences for having clear meaning.

Materials and Methods:

Data collection - 1. Was the PZQ administered after hematuria detection or after urine filtration or both? 2. If the first then why not after the urine filtration, which is more accurate detection method. 3. Provide details of the diagnostic procedures as readers are not aware about the parent study. 4. Discuss the aspects of wrongfully tested described in the flow diagram.

Analysis - This section is too long. Please try to use some sub-headings before the paragraphs.

Results:

Line 213 - Should be different.

Discussion:

Line 237-245 - The non significance between treated vs non-treated children, may be due to: 1. likely the control group had infection as hematuria is being observed one time to assign them into groups. No sensitive and effective diagnostic test was not used. So, likely missed infection and thus treatment.  2. The reinfection, recurring infection, and remaining infection are common for Schistosomiasis. Was any of these aspects taken into consideration.  3. Was infection with other soil transmitted helminths taken into consideration, which can impact under body weight, stunting, and wasting.

Line 250-253 - This statement is conflicting as the notion of under-nutrition is long-term, which may not be impacted by short-term lack of eating which may not be exhibited most of the treated children. Was this aspect being recorded to draw this conclusion? If not then please disregard.

Line 257 -260 - Other than longer follow-up, may be multiple round of MDA via PZQ can improve the outcome. Also, other social determinants of health factors, such as food insecurity, availability and access to health foods, financial insecurity can play as an important co-factor and should be considered.

Line 263-268 - Put these into context with the above-mentioned points.

Line 268-271 - This is a serious concern, as the practice is that the consumption of PZQ happens in the presence of CHWs. There is well-established literature to support the practice. So, the presented number of children part of the treated group can be obscured.

Line 299-308 - This section is talking about social determinants of health issues that has been mentioned earlier. These are very strong argument that I agree with. Please provide few more references to bolster this section.

Line 309-316 - These two sections have lot of similarity. please delete the last section and keep the conclusion.

Author Response

Dear Reviewer 

Thank you for your comments and suggestions. We appreciate the opportunity to address your comments in further improving our paper so that it can be rendered acceptable for publication in Pathogens. Below you will find a point-by-point explanation as to how we have addressed your specific feedback.  We look forward to hearing your decision.

Reviewer 1

Introduction

  1. Line 67-70: This sentence needs rewriting or breaking it into two sentences for having clear meaning.
  • Thank you for pointing out the need to elucidate this sentence. This sentence has been broken up into two. Please see the changes under introduction on page 2, paragraph 3.

The revised sentence reads: “This study, therefore, provides evidence on how quickly undernutrition can be reversed among children with S. haematobium treated with PZQ. It is critical to understand how fast undernutrition can be addressed in children infected with in S. haematobium as prolonged undernutrition increases the likelihood of permanent developmental deficits and reduces the chance of  achieving developmental “catch-up” later in life [14, 15].”

Materials and Methods

  1. Data collection – 1. Was the PZQ administered after hematuria detection or after urine filtration or both?
  • Children who were positive for hematuria and/or haematobium egg (s) were treated. This has been clarified. The revised sentences about the treatment reads: “To expedite treatment in light of the sheer volume of children being analyzed, those whose urine samples tested positive for hematuria were provided with praziquantel (Bermoxel by Medochemie, Limassol, Cyprus) at a single dose of 40 mg/kg… Efforts were made to ensure that children who tested negative for hematuria but who tested positive using urine filtration microscopy were promptly treated with PZQ.” Please see section 2.3. on page 3, paragraphs 2 – 3.

  1. If the first then why not after the urine filtration, which is more accurate detection method.
  • Treatment was done after hematuria and/or urine filtration microscopy. Please see our response above.

  1. Provide details of the diagnostic procedures as readers are not aware about the parent study.

  • Details on the diagnostic procedures have been added (see section 2.3. on page 3, paragraphs 2 – 3).

The revised text reads: “The instructions by the manufacturer were followed to check the presence of hematuria using Combur 10 Test reagent strips. In brief, the test strip was immersed in the urine sample for about two seconds, then removed and left on the bench for approximately one minute. The blood area on the test strip was then compared to the corresponding reference color fields. The strip test results were interpreted and recorded as zero (negative), ± (trace), + (weak), ++ (moderate), and +++ (strong), indicating the presence of erythrocytes per μL of urine.

All samples underwent confirmatory testing for S. haematobium eggs using standard urine filtration microscopy in accordance with the parent study [16] by which a 10 mL aliquot of each urine sample was filtered through a polycarbonate membrane and then analyzed for S. haematobium eggs under the microscope in the lab.”

  1. Discuss the aspects of wrongfully tested described in the flow diagram.

  • We have added an explanation as to why those children received the incorrect treatment (see section 2.3. on page 3, paragraph 3). The added text reads: “ Of the 245 children who were confirmed positive for haematobium at baseline, 52 were excluded from the study because they tested negative for hematuria in the field and were therefore not treated until later. Likewise, of the 721 children who were confirmed negative for S. haematobium eggs, 91 were excluded because they tested positive for hematuria in the field and were therefore given praziquantel.” To reinforce this clarification, we have also added footnotes under the flow chart on page 4.

  1. Analysis - This section is too long. Please try to use some sub-headings before the paragraphs
  • Sub-headings have been added in the data analysis section. We have added three subheadings in the data analysis section including

2.4.1. Determining nutritional status

2.4.2. Association of S. haematobium infection with undernutrition at baseline

2.4.3. Effect of praziquantel treatment on nutrition status

        Results

  1. Line 213 - Should be different.
  • This typo has been corrected. See section 3.3. on page 7, paragraph 2.

       Discussion

  1. Line 237-245 - The non significance between treated vs non-treated children, may be due to: 1. likely the control group had infection as hematuria is being observed one time to assign them into groups. No sensitive and effective diagnostic test was not used. So, likely missed infection and thus treatment.
  • Thank you for bringing this to our attention. Assignment of the study participants to the treatment vs control groups was made based on results of the urine filtration microscopy. We have now clarified this in the manuscript. The added text reads: “…by which a 10 mL aliquot of each urine sample was filtered through a polycarbonate membrane and then analyzed for haematobium eggs under the microscope in the lab, the results of which were used to group children into treated (i.e. urine samples tested positive for S. haematobium egg) and control (i.e. urine samples tested negative for S. haematobium egg) groups.” Please see section 2.3. on page 3, paragraph 3. We hope this revision in the methods section will address your concern in the discussion.

  1. The reinfection, recurring infection, and remaining infection are common for Schistosomiasis. Was any of these aspects taken into consideration. 3. Was infection with other soil transmitted helminths taken into consideration, which can impact under body weight, stunting, and wasting.
  • Thank you for bringing this issue to our attention. We have emphasized these points as limitations to our study in the discussion on page 8, paragraph 5. The added text in the limitation paragraph in the discussion reads: “…we were unable to collect richer data that would have enabled us to control for other biasing factors, including a history of past infection, co-infection with other parasites and soil-transmitted helminths, exposure to bodies of water during the follow-up period which may cause rapid reinfection, and other social determinants of under nutrition, such as food insecurity, availability and access to health foods and financial insecurity.”

  1. Line 250-253 - This statement is conflicting as the notion of under-nutrition is long-term, which may not be impacted by short-term lack of eating which may not be exhibited most of the treated children. Was this aspect being recorded to draw this conclusion? If not then please disregard.
  • Thank you for your insight on this matter. We have removed from the discussion the sentence which reads: “Although it is plausible that praziquantel’s ability to kill schistosomes can lead to im-proved nutritional status by removing a parasite that consumes the host’s metabolites, it should be noted that the drug’s side effects, including abdominal pain, nausea and head-ache [19], could, in effect, lead to undernutrition in the short-term as children shun eating in the presence of such symptoms.”

  1. Line 257 -260 - Other than longer follow-up, may be multiple round of MDA via PZQ can improve the outcome. Also, other social determinants of health factors, such as food insecurity, availability and access to health foods, financial insecurity can play as an important co-factor and should be considered.
  • Thank you for your suggestions. We have suggested the importance of considering multiple round of treatment and other social determinants of undernutrition, such as food insecurity, availability and access to health foods, financial insecurity when evaluating the effect of treatment with PZQ on nutritional status among children in future studies. The revised sentences are as follows –

“This underscores the need to conduct future studies with longer follow-up periods and maybe with repeated PZQ treatment.” (see discussion on page 8, paragraph 4)

“Nonetheless, because of the context of this research setting as well as resource limitations, we were unable to collect richer data that would have enabled us to control for other biasing factors, including a history of past infection, co-infection with other parasites and soil-transmitted helminths, exposure to bodies of water during the follow-up period which may cause rapid reinfection, and other social determinants of under nutrition, such as food insecurity, availability and access to health foods and financial insecurity.” (see discussion on page 8, paragraph 5)

  1. Line 263-268 - Put these into context with the above-mentioned points.
  • Thank you. We would like the reviewers to review our response to the comments above

  1. Line 268-271 - This is a serious concern, as the practice is that the consumption of PZQ happens in the presence of CHWs. There is well-established literature to support the practice. So, the presented number of children part of the treated group can be obscured.
  • We share the concern and expanded the sentence discussing the nature of uptake of the drug as a limitation (see discussion on page 9, paragraph 1). It reads: “Given that the praziquantel was provided to parents with instructions for administration to their infected child[ren] and that during the follow-up survey the researchers were unable to ascertain drug consumption due to logistical constraints, only an intent-to-treat analysis was performed. As such, the nondifferential misclassification that may have resulted would have likely biased the results of this study toward the null. As such, the nondifferential misclassification that may have resulted would have likely biased the results of this study toward the null.”

  1. Line 299-308 - This section is talking about social determinants of health issues that has been mentioned earlier. These are very strong argument that I agree with. Please provide few more references to bolster this section.
  • We have added references to support our argument on the social and environmental determinants of undernutrition [references # 28-31]. (please see the discussion on page 9, paragraph 4).

  1. Line 309-316 - These two sections have lot of similarity. please delete the last section and keep the conclusion.
  • We have deleted the last paragraph of the discussion as suggested.

Reviewer 2 Report

Comments and Suggestions for Authors

The authors wanted to investigate the impact of Praziquantel on the nutritional status of the children infected with Schistosoma Haematobium.

1. The introduction is too short.

2. Authors have reported that there is no change observed and suggested that more studies need to be conducted. I am unsure why they want to submit this manuscript if more study is needed. Authors should be clear when reporting their findings. If something is missing please add that to the limitation subsection.

3. Was there any other diagnostic marker used to check the treatment outcomes of PZQ in the treated children? Authors are requested to include any clinical test data. This is crucial information as it will help validate that the PZQ post-1-month treatment does not affect the nutritional status of the children.

4. The conclusion can be rewritten to capture the whole story.

Author Response

Dear Reviewer,

Thank you for your comments and suggestions. We appreciate the opportunity to address your comments in further improving our paper so that it can be rendered acceptable for publication in Pathogens. Below you will find a point-by-point explanation as to how we have addressed your specific feedback.  We look forward to hearing your decision.

  1. The introduction is too short.
  • We have expanded the introduction a little bit. Please see the added text in the introduction in highlights.
  1. Authors have reported that there is no change observed and suggested that more studies need to be conducted. I am unsure why they want to submit this manuscript if more study is needed. Authors should be clear when reporting their findings. If something is missing please add that to the limitation subsection.
  • Thank you for the suggestion. The current study presents findings with significant public health and research implications, making it worthy of publication in Pathogens. These findings include the association between haematobium infection and undernutrition, as well as a reduction in the prevalence of underweight among children infected with S. haematobium one month after treatment with praziquantel (PZQ). However, no changes were observed in BAZ, WAZ, height-for-age z-scores (HAZ), or MUAC scores one month post-treatment.

As anticipated, the study has certain limitations. Therefore, we have proposed additional studies to address these limitations and to confirm the long-term effects of S. haematobium infection on undernutrition, as well as to determine whether S. haematobium-related undernutrition is reversible.

Moreover, we have further reviewed the manuscript, made edits to improve clarity, and expanded the section discussing the study’s limitations to thoroughly address all identified weaknesses.

  1. Was there any other diagnostic marker used to check the treatment outcomes of PZQ in the treated children? Authors are requested to include any clinical test data. This is crucial information as it will help validate that the PZQ post-1-month treatment does not affect the nutritional status of the children.
  • We acknowledge that incorporating clinical data could have complemented the anthropometric parameters in assessing the nutritional status of the study participants. Unfortunately, clinical data related to nutritional status were not collected in this study, which has been discussed as one of its limitations. The added text reads: “Likewise, we were unable to measure more facets of nutritional status, including those that can only be measured using clinical/laboratory testing, including blood iron levels, blood glucose levels, proteinuria, skin thickness, among others, as were done in other studies [7, 23].” (Please see discussion on page 9, paragraph 1)
  1. The conclusion can be rewritten to capture the whole story.
  • We have revised the conclusion both in the abstract and in the discussion.

Conclusion in the abstract reads “In conclusion, children infected with S. haematobium-infected are likely to suffer from undernutrition; however, a single dose PZQ therapy may not improve their nutritional status within one month of treatment. Future studies could have longer follow-up periods to better estimate the drug’s effect on nutrition.” (See abstract on page 1)

Conclusion in the discussion reads “Urogenital schistosomiasis may cause underweight and wasting in school-aged children. However, treating the infection with a single dose of praziquantel does not ap-pear to significantly improve anthropometric nutritional parameters within one month of treatment. Future studies should consider monitoring children over longer periods, incorporating repeated praziquantel treatments to better understand the potential for re-versing S. haematobium-related undernutrition. Controlling for other social and environ-mental factors influencing nutrition will also be essential during such studies.” (See conclusion on page 9, paragraph 5).

Round 2

Reviewer 1 Report

Comments and Suggestions for Authors

All comments were addressed adequately. No further comments. 

Reviewer 2 Report

Comments and Suggestions for Authors

The current manuscript discusses the "Effect of Praziquantel Treatment on the Nutritional Status of Children Infected with Schistosoma haematobium". The authors have systematically answered all the questions. Sharing of reference 11 in the introduction clarifies their perspective and information shared. I think this manuscript can be accepted.